

# Characterisation of the melting layer variability in an Alpine valley based on polarimetric X-band radar scans

Floor van den Heuvel[1,2], Marco Gabella[2], Urs Germann[2], and Alexis Berne[1]

[1]Ecole Polytechnique Fédérale de Lausanne (EPFL), Lausanne, Switzerland
[2]Swiss Federal Office of Meteorology and Climatology (MeteoSwiss), Locarno-Monti, Switzerland

**Correspondence:** Alexis Berne (alexis.berne@epfl.ch)

**Abstract.** The melting layer designates the transition region from solid to liquid precipitation, and is a typical feature of the vertical structure of stratiform precipitation. As it is characterised by a well-known signature in polarimetric radar variables, it can be identified by automatic detection algorithms. Though often assumed to be uniform in space and time for applications such as vertical profile correction, the spatial variability of the melting layer remains poorly documented. This work undertakes

to characterise and quantify the spatial and temporal variability of the melting layer using a method based on the Fourier transform, which is applied to high resolution X-band polarimetric radar data from two measurement campaigns in Switzerland. It is first demonstrated that the proposed method can accurately and concisely describe the spatial variability of the melting layer and may therefore be used as a tool for comparison. The method is then used to characterise the melting layer variability in summer precipitation on the relatively flat Swiss plateau and in winter precipitation in a large inner Alpine valley (the Rhone

valley in the Swiss Alps). Results indicate a higher contribution of smaller spatial scales to the total melting layer variability in the case of the Alpine environment. The same method is also applied on data from vertical scans in order to study the temporal variability of the melting layer. The variability in space and time is then compared to investigate the spatio-temporal coherence of the melting layer variability in the two study areas, which was found to be more consistent with the assumption of pure advection for the case of the plateau.

**1   Introduction**

Quantitative precipitation estimation (QPE) with radar in complex terrain such as the Alps, is complicated by many factors amongst which partial and total beam shielding by terrain, the influence of orography on the dynamics and microphysics of precipitation as well as the shallow depth of precipitation during cold seasons (Germann and Joss, 2004; Roe, 2005; Houze, 2012; Colle et al., 2013). In order to avoid the problem of shielding, radar measurements are often collected at higher ele-

vations. The measurements made aloft are then usually extrapolated to the ground level to compensate for the lack of direct visibility with the radar (Joss and Pittini, 1991; Joss and Lee, 1995; Andrieu and Creutin, 1995; Vignal et al., 1999; Germann and Joss, 2002; Gray et al., 2002; Vignal et al., 2000; Zhang and Qi, 2010; Kirstetter et al., 2013). These extrapolated values are commonly corrected with the vertical profile of reflectivity (VPR) which represents the vertical change in the radar reflectivity measurement due to changes in size, phase and fallspeed of hydrometeors. Many VPR correction techniques used





operationally, are based on mean VPRs extracted in well-visible regions of the radar (Koistinen, 1991; Joss and Lee, 1995; Germann and Joss, 2001, 2002). Other broad categories include climatological VPRs (Joss and Pittini, 1991; Joss and Lee, 1995; Gray et al., 2002), inverse VPRs (Andrieu and Creutin, 1995; Andrieu et al., 1995; Vignal et al., 1999; Vignal and Krajewski, 2001) and model derived VPRs (Mittermaier and Illingworth, 2003; Zhang and Qi, 2010; Kirstetter et al., 2013). A typical

feature of VPRs in stratiform precipitation is the melting layer (ML) or bright band signature which designates a transition region from solid precipitation to liquid precipitation. It is characterised by a high horizontal reflectivity factor ($Z_H$) due to the increase in effective dielectric constant as solid hydrometeors are coated by a thin layer of water, as well as a decrease in the copolar correlation coefficient ($\rho_{HV}$) due to the presence of heterogeneous hydrometeor types (Battan, 1973; Zrnic et al., 1993; Fabry and Zawadzki, 1995; Brandes and Ikeda, 2004). Other polarimetric signatures in the melting layer include an increase

in differential reflectivity ($Z_{DR}$) which is usually smaller in the solid phase and higher in the liquid phase (Doviak and Zrnić, 1993) and large linear depolarisation ratio (LDR) values which may be related to broader distributions of canting angles due to increased spinning of the hydrometeors (Brandes and Ikeda, 2004). However, layers with single pristine crystals such as dendrites which are often present above the melting layer may cause a similar increase in $Z_{DR}$ values (Matrosov et al., 2007), which is why many melting layer detection algorithms are based on $Z_H$ and $\rho_{HV}$ or LDR measurements.

Algorithms for QPE and VPR extraction often assume that the melting layer is spatially and temporally homogeneous, however as the VPR shape is dependent on microphysical processes such as riming and aggregation (Fabry and Zawadzki, 1995; Bell, 2000; Roe, 2005; Stoelinga et al., 2013), as well as on the vertical profiles of temperature and relative humidity (Matsuo and Sasyo, 1981; Stoelinga et al., 2013), this assumption may not necessarily hold for events with rain-snow transitions or in an orographic context (Boodoo et al., 2010; Campbell and Steenburgh, 2014). Multiple studies have shown that the melting layer

can dip a few hundred metres downwards in the proximity of terrain (Lumb, 1983; Marwitz, 1983; Houze and Medina, 2005; Medina et al., 2005; Marigo et al., 2008; Stoelinga et al., 2013). And the melting layer depth, for example, may vary with a factor of three depending on snowflake density (Matsuo and Sasyo, 1981; Stoelinga et al., 2013). Indeed, based on observational data, Wolfensberger et al. (2016) found significant dependencies between melting layer thickness and the presence of rimed particles above the melting layer. The authors related this to the longer distances travelled by rimed particles before

complete melting due to the higher densities and fall velocities of these types of hydrometeors. Other important identified factors explaining the variability of the melting layer included the reflectivity gradient in the solid phase and copolar correlation coefficient values inside the melting layer (Wolfensberger et al., 2016).

As a result, individual VPRs at the near and far range can be highly variable with respect to the average VPR even in non-mountainous terrain (Jordan et al., 2000; Bellon et al., 2005; Berne et al., 2004). Though the variability of the melting layer and

the freezing level height has been extensively studied at seasonal and large spatial (Harris et al., 2000; Thurai et al., 2003; Das et al., 2011; Rudolph and Friedrich, 2013) and temporal (Fabry and Zawadzki, 1995) scales, information on and quantification of the small scale spatial variability remains relatively sparse. Das et al. (2011) for example, studied the seasonal variability of the melting layer height at two different locations in India and found that the higher latitude location showed more monthly variability. For Mediterranean precipitation Berne et al. (2004) found that point-scale VPRs have a spatial representativeness

which is limited to 6 km distances from the radar for 15 minutes integration times. Whereas Fabry et al. (1994b) reported a





melting layer height change of 1500 metres within 3 hours in Montreal (Canada), Cluckie et al. (2000) found no more than 600 meters deviation in the melting layer height in the region of Middle Wallop (England) and this only in conditions with significant convection. However, none of these studies were conducted in an Alpine environment.

Mohymont and Delobbe (2008) first evaluated the ability of variograms to assess the spatial variability of the vertical profile of reflectivity in volumetric radar data in Belgium. They found that the variations of variograms of VPRs up to 200 km from the radar were caused by the non-uniform nature of the vertical structure. Variograms of reflectivity were also used by Germann and Joss (2001) to quantify the spatial variation of Alpine precipitation, which was found to be considerable for various types (convective and stratiform) of precipitation. Though the differences in reflectivity at all spatial ranges were lower for stratiform precipitation than for convective precipitation, the authors also found that the variation was weaker above the melting layer than below, indicating that the variation in reflectivity aloft can not fully explain the variation observed at ground levels. The vertical structure of radar-observed precipitation in Switzerland was also studied by Rudolph and Friedrich (2013). Based on characteristic seasonal patterns in the vertical structure, the authors could perform a seasonal classification of storms. The authors further related the vertical structure to dynamic and thermodynamic environmental parameters, showing that the radar-observed vertical structure of precipitation in the vicinity of Locarno, Switzerland, is correlated with synoptic patterns, integrated water vapour flux, atmospheric stability and the vertical profiles of temperature, moisture and wind (Rudolph and Friedrich, 2014). They could predict the vertical storm structure type with reasonable accuracy based on these parameters.

To increase the availability of radar information, MeteoSwiss has recently extended its weather radar network with two polarimetric C-band radars at high altitude locations (∼3000 m asl) in the Swiss Alps (Germann et al., 2015). This poses a new challenge of the applicability of existing vertical profile correction techniques to these high-altitude measurements but also provides new opportunities to use polarimetric radar variables for the improvement of QPE in these regions. Thus, information on, and the quantification of the spatial variability of the vertical structure of polarimetric radar variables in mountainous terrain is an important first step to the improvement of QPE and the estimation of its uncertainty in the Swiss Alps. Two measurement campaigns were conducted in very different though highly representative regions in Switzerland in order to study the structure and variability of precipitation. The use of mobile radars allowed to study the lower part of the troposphere in places of interest and with reduced visibility from the operational C-band radars. The melting layer detection algorithm designed by Wolfensberger et al. (2016) has been applied to the RHI (range-height indicator) scans from these campaigns to extract information on various characteristics of the melting layer. This paper seeks to quantify and compare the spatial variability of these melting layer characteristics at different spatial scales using a method based on the Fourier transform. The paper is structured as follows; section 2 briefly presents the datasets and study areas, section 3 describes the pre-processing and the method for the quantification of the spatial variability, section 4 presents an evaluation of the method as well as the results and discussion of the melting layer statistics from the two study areas and section 6 contains the main conclusions.




## 2   Data

Most of the RHI scans used in this work were performed by the EPFL-LTE X-band Doppler dual polarization radar (MXPol)
which was deployed in two distinct areas in Switzerland. First within the context of the PaRaDIso (PAyerne RADar and ISO-
topes) campaign on the Swiss plateau in Payerne from the end of March to the beginning of July 2014 (Fig. 1), where it was
co-located with another X-band radar (DX50). This location was of particular interest because it represents the climate condi-
tions of a large, and most densely populated part of Switzerland and because of the presence of various other (disdrometers,
radars, profilers, sounding) instruments in the area. MXPol was then deployed at the ground level (460 m asl) near Martigny
in the main valley of the Swiss Alps for the 2016-2017 winter season. This area has the advantage of being both a very deep
and long valley, thus providing both relatively good visibility with the mobile X-band radar and the possibility of studying
precipitation characteristics in an Alpine valley.

[FIGURE 1 about here]

### 2.1   PaRaDIso campaign

The MXPol radar operated in Payerne from the 21$^{st}$ of March until the 14$^{th}$ of May within the context of the PaRaDIso cam-
paign (hereafter Payerne campaign). It was offline for maintenance for one week between the 21$^{st}$ and the 28$^{th}$ of April, but no
significant precipitation events occurred during this period. Datasets from this campaign include RHI scans from the MXPol
radar as well as those from a similar X-band radar (DX50), located approximately 3.7 km away from MXPol. The scanning
strategy of the MXPol radar during this campaign consisted of two RHIs (one in the direction of the DX50 radar) a Plan Po-
sition Indicator (PPI) scan at 5° elevation and a vertical PPI scan. The DX50 performed 3 RHI's (one in the direction of the
MXPol), 3 PPI's and also a vertically pointing scan. The scan strategies of both radars were repeated every 5 minutes. There are
a total of 10 significant events for which data were recorded by both radars, representing over 170 hours of precipitation during
which a total of 61.5 mm of rain was recorded in the nearby rain gauge. In addition to this, there were 4 medium intensity
events, which represent over 29 hours of data and for which 5.4 mm of rain was recorded in the gauge. Data from both radars is
also available for 5 low intensity events. All these events constituted the basis for the selection of the radar data for continuous
and sufficiently long melting layers. About 460 RHI scans per azimuth were retained, each event containing at least 20 RHI
scans. More information on this campaign can also be found in (Figueras i Ventura et al., 2015).

[FIGURE 2 about here]

The left most panel in Fig. 2 shows the average number of days with precipitation in Payerne for the period of March until
May 2014 compared to the average conditions at that location for the same period. March 2014 for example, was relatively dry,
whereas May was slightly wetter than usual. The middle panel shows the number of scans with a detected melting layer for
each month; there is a clear over-representation for the month of May, mainly due to the fact that this is a rainier month both in
terms of number of rainy days (left panel) and precipitation amounts (right panel). The climatogram in the right panel indicates





average monthly precipitation sums (blue) and temperatures (green, with the quartiles in different shades) as compared with the measurements over the period of the Payerne campaign. The observed precipitation sums have been subdivided into the contributions of four classes of precipitation intensity. The total precipitation sums for the month of May did not exceed the climatological average, even though the number of wet days was higher. The average observed monthly temperatures are

represented with red dots and are well within the climatological limits.

## 2.2 Valais 2016-2017 campaign

The set-up of a mobile X-band radar near Martigny scanning under the operational MeteoSwiss C-band radar located at Plaine Morte (∼40 km distance) was specifically intended to study precipitation in an orographic context and in wintertime, which are the conditions known to be most challenging to the quantitative estimation of precipitation in Switzerland (Germann and Joss,

2004; Koistinen et al., 2004; Germann et al., 2006; Montopoli et al., 2017). The MXPol radar measured precipitation in the Valais from the 3$^{rd}$ of November 2016 to the 24$^{th}$ of May 2017. From this period, only the measurements between November 2016 and March 2017 are taken into account in order to restrict the analysis as much as possible to winter precipitation. The main characteristics of the MXPol radar and the principal scanning strategy performed during the Valais campaign are given in Table 1. Because the radar was located in a deep valley, no non-vertical PPI scans were performed. The scan strategy changed

once early during the campaign; before that, the radar performed 3 RHI scans in Dual Pulse Pair (DPP) mode in the main axis of the valley and one vertical PPI scan for the $Z_{DR}$ calibration. The scanning strategy described in Table 1 turned for the remainder of the campaign and performed one RHI scan in the direction of the Plaine Morte (47°), one hemispheric RHI in the axis of the main valley (54°), one RHI in Fast Fourier Transform (FFT) mode in the direction of Pierre Avoi mountain (90°), one RHI which made a cross section of the main valley (147°) and the vertical $Z_{DR}$ calibration scan in FFT mode. Both FFT

mode scans were performed at 30 metres resolution and the DPP mode scans were performed with 75 metres resolution.

[TABLE 1 about here]

During the considered time period, a total of 403 hours of precipitation were recorded at the ground station in Martigny. Roughly 16 precipitation events could be identified on the basis of more or less continuous precipitation and similar synoptic

conditions, representing a total of 324 hours of precipitation. These 16 events constituted the basis for the selection of the radar data for continuous and sufficiently long melting layers. Eventually, 1651 RHI scans per azimuth direction were retained, with each event containing at least 50 RHI scans.

[FIGURE 3 about here]

The left panel in Fig. 3 shows the average number of days with precipitation in Sion (∼20 km distance from MXPol) compared with the observed number of days with precipitation at this location during the Valais campaign. While December and January were unusually dry, the other months were slightly wetter. The middle panel in Fig. 3 shows the number of scans





with a detected melting layer for each month. The month of March is slightly over-represented due to the higher number of wet days and the higher precipitation sums for that month (right panel). The climatogram in the right panel indicates average monthly precipitation sums (blue) and temperatures (green, with the quartiles in different shades) as compared with the measurements over the period of the Valais campaign. The observed precipitation sums have been subdivided into the

contributions of four classes of precipitation intensity. Precipitation sums for the months of May and November exceeded by far the climatological average, while February and March were relatively warm.

## 3   Methodology

### 3.1   Pre-processing

MXPol applies an automatic Doppler filter on non-vertical scans, and all the radar data from the measurement campaigns

have been further subjected to the same pre-processing routine including a filtering of the data based on the signal-to-noise ratio threshold of 5 dB (10 dB for all phase based data)[1] and a copolar correlation coefficient ($\rho_{hv}$) threshold of 0.6. Horizontal reflectivity ($Z_H$) and differential reflectivity ($Z_{DR}$) have been corrected for attenuation in rain using the constrained method by Testud et al. (2001). For the Payerne data a single $Z_{DR}$ calibration coefficient, based on observations from previous campaigns, was applied. For the Valais campaign, due to increased variability of the $Z_{DR}$ likely related to higher temperature variations

on site, the calibration coefficient was updated more frequently and calculated based on data from the solid phase similar to the approach described by Dixon et al. (2017). The specific differential phase ($K_{dp}$) was estimated from the total differential phase shift ($\Psi_{dp}$) using the multistep method described by Vulpiani et al. (2012)[2].

All the RHI scans from the 16 identified precipitation events were subjected to the melting layer detection algorithm developed by Wolfensberger et al. (2016), which was run using the ARM Radar Toolkit (Py-ART) (Helmus and Collis, 2016). The

algorithm uses the gradients of reflectivity and copolar correlation coefficient to detect the melting layer top and bottom; more information on the algorithm can be found in Wolfensberger et al. (2016). Before applying the algorithm, the lowest elevation angles and the furthest gates of the scans were cut off to avoid contamination from ground clutter and mountains. In addition to this, the RHIs were subjected to a texture based clutter filter from the pyART toolbox. In order to limit the effects of beam broadening, the melting layer detection algorithm has been set to detect up to a maximum distance of 10 km from the radar,

holes in the detected melting layer tops and bottoms were interpolated up to a maximum length of 1500 metres such as to obtain continuous, non interrupted data series. Otherwise, the default settings found to be optimal and described by Wolfensberger et al. (2016) were used.

### 3.2   Power spectra

Spectral analysis, similarly to variograms, is a frequently used tool to study the second-order properties of a process. In me-

teorology, it has been used in fields ranging from boundary layer meteorology (Van der Hoven, 1957; Stull, 1988), radar

---

[1] for the Payerne data thresholds were 0 dB and 5 dB respectively

[2] Kalman-filtering was used for the Payerne data





observations of turbulence (Crane, 1980), the analysis of the spatial representativeness of precipitation forecasts (Harris et al., 2001) to probabilistic nowcasting (Bowler et al., 2006; Nerini et al., 2017). Spectral analysis has also been applied to reveal the scaling behaviour of precipitation over both temporal (Fraedrich and Larnder, 1993) and spatial (Mandapaka et al., 2009) scales as well as the correlation of these (Rysman et al., 2013). And the spectral slope (or $\beta$ slope) values have been found to be

dependent on the underlying meteorological processes; convective rain processes for example, having steeper spectral slopes than stratiform ones (Purdy et al., 2001; Nykanen and Harris, 2003; Nykanen, 2008). To our knowledge, spectral analysis has not been used to study the melting layer variability.

From the output of the melting layer detection algorithm various variables have been extracted and calculated such as the

heights of the top and bottom of the melting layer, the thickness or depth of the melting layer and the height of the reflectivity peak within the melting layer. Figure 4 shows the melting layer detection algorithm output with the corresponding extracted and derived variables for an idealised vertical profile.

[FIGURE 4 about here]

Before subjecting the data to the Fourier transform, it was conditioned following the indications in Stull (1988). More specifically, the melting layer variables were first de-trended and then tapered to avoid red noise and leakage. The tapering was done with a bell taper for which the window weight is given by:

$$\boldsymbol{W}(k) = \begin{cases} \sin^2(5\pi k/\boldsymbol{N}) & \text{for } 0 \le k \le 0.1\boldsymbol{N} \\ 1 & \text{elsewhere} \\ \sin^2(5\pi k/\boldsymbol{N}) & \text{for } 0.9\boldsymbol{N} \le k \le \boldsymbol{N} \end{cases} \tag{1}$$

The variables were then subjected to a one dimensional Fast Fourier transform, such that for an array of melting layer tops

$\boldsymbol{m}$ of length $N$ where $\boldsymbol{A}(k)$ represents a single data point:

$$\boldsymbol{F_A}(n) = \sum_{k=0}^{N-1} \frac{\boldsymbol{A}(k)}{N} * \exp^{-i2\pi kn/N} \tag{2}$$

Since only continuous melting layers were taken for the analysis, padding (filling the gaps with artificial data points) was not necessary. The Fourier transform results in $N$ coefficients of $\boldsymbol{F_m}(n)$ each having a real part and an imaginary part ($\boldsymbol{F_m}(n) = \boldsymbol{F}_{\mathbb{Re}}(n) + i\boldsymbol{F}_{\mathbb{Im}}(n)$). The square of the norm of the complex Fourier transform for any frequency $n$ is:

$$|\boldsymbol{F_A}(n)|^2 = \boldsymbol{F}_{\mathbb{Re}}^2(n) + \boldsymbol{F}_{\mathbb{Im}}^2(n) \tag{3}$$





The fraction of variance explained by each component $n$ can be derived by summing the square of the norm ($|\boldsymbol{F_A}(n)|^2$) over $n = 1$ to $N - 1$ ($n = 0$ is excluded because $|\boldsymbol{F_A}(0)|$ is the mean value) resulting in the total variance of the original series:

$$\sigma_{\boldsymbol{A}}^2 = \frac{1}{N} \sum_{k=0}^{N-1} (\boldsymbol{A}_k - \overline{\boldsymbol{A}})^2 = \sum_{n=1}^{N-1} |\boldsymbol{F_A}(n)|^2 \tag{4}$$

The fraction of variance explained by component $n$ is then given by dividing the square of the norm by the total variance:

$$\frac{|\boldsymbol{F_A}(n)|^2}{\sigma_{\boldsymbol{A}}^2} \tag{5}$$

Note that this is different from the discrete spectral density, which is calculated as:

$$\boldsymbol{E_A}(n) = 2 \times |\boldsymbol{F_A}(n)|^2, \quad \text{for } n = 1 \text{ to } n_f \text{ with } N = \text{odd}$$
$$\boldsymbol{E_A}(n) = 2 \times |\boldsymbol{F_A}(n)|^2, \quad \text{for } n = 1 \text{ to } (n_f - 1) \text{ with } N = \text{even}$$
$$\boldsymbol{E_A}(n) = |\boldsymbol{F_A}(n)|^2, \quad \text{at } n_f$$

10    And from which the spectral energy density can be approximated:

$$\boldsymbol{S_A}(n) = \frac{\boldsymbol{E_A}(n)}{\Delta n} \tag{6}$$

For a physical process which is scale-invariant in the space or time domain, the power spectrum $\boldsymbol{S}(f)$ approaches the power law such that:

$$\boldsymbol{S}(f) \propto f^{-\beta} \tag{7}$$

15    The $\beta$ value, which is more commonly used in literature, can be found using linear regression of the log-log plot of $\boldsymbol{S}(f)$ and $f$ (Davis et al., 1996; Harris et al., 1997; Purdy et al., 2001). Since autocorrelation is the inverse Fourier transform of the power spectral density, the $\beta$ value shows how fast the autocorrelation decreases with lag. A higher $\beta$ value corresponds to a steep spectral slope and thus a highly correlated process for which the contribution to the signal of low-frequency components is large in relation to the contribution from high-frequency components. A low $\beta$ value on the other hand, corresponds to a low
20    spectral slope, fast decreasing autocorrelation and a higher relative contribution from high-frequency components. Theoretically, the $\beta$ value is 0 for white noise and 2 for pure Brownian noise.

Spectral slopes were also calculated within the context of this study. However, fitting to single spectra of a single realisation of the signal resulted in large uncertainties for the spectral slope values. Whereas the averaging of the signals (i.e. for two-dimensional data typically some azimuthal averaging is performed) has the effect of smoothing the power in the low-frequency





components leading to spectral slopes which give very little information on the variability of the signal at the larger spatial lags. Therefore the analysis in this study is based on the fraction of variance explained by component. It has the advantage of summarising the information on the spatial variability into a few components, showing the relative amount of variance explained by each spatial or temporal lag. It thus allows for the comparison of individual melting layers as well as the aggregation of

data, while preserving information on the uncertainty when presented, for example, in box plots. Still, as Eq. (6) and Eq. (7) demonstrate, the fractions of variance explained by component are related to the spectral slopes, thus facilitating comparison with existing literature.

The effects of the described steps for the conditioning of the data on the fractions of variance explained by component were monitored, and are briefly discussed in the following section.

## 10   4   Results

In the subsections below the extent to which the method can describe the individual melting layers, and how the fractions of variance explained by component are affected by the conditioning of the data are treated first. Then the results from the study areas on the Swiss plateau and in the Swiss Alps are compared, first based on general statistics of the melting layer and then based on the fractions of variance explained by component for the spatial analysis. Finally, the spatio-temporal coherence of

the melting layer variability is assessed for the two sites.

### 4.1   Evaluation of the method

In order to evaluate the extent to which the method can describe the melting layer variability and how many components are needed, the performance on individual melting layers was analysed. Figure 5 shows a selection of three melting layer tops from the Payerne campaign and two melting layers tops from the Valais campaign. The blue line is an artificial melting layer created

by adding white noise to a linear trend. For the fraction of variance explained by component (middle panel), only the first ten spatial frequencies are given (here represented in wavelength ($1/f$)) for representational purposes. The bottom panel gives the cumulated sum of the fractions of variance explained by component from the 5000 m distance lag down to the Nyquist frequency (two times the resolution of the data). Since the spectra have not been folded back such as for the calculation of the discrete spectral density, the maximum fraction of explained variance is at 0.5. As a scaling break can be observed around the

500 m wavelength at minimum (indicated with a vertical dashed line) and since the most important differences between the melting layers are concentrated in the first few components, higher frequencies are considered to approach the noise related either to the melting layer detection algorithm or the resolution of the data. This is also close to the 750 m distance lag found by Fabry et al. (1994a) as the separation between variability due to measurement noise and weather. Moreover, as the bottom panel in Fig. 5 as well as calculations over the entire dataset indicate, at this distance the cumulated explained variance is 50%

or more of the total variance in most of the melting layer tops.

[FIGURE 5 about here]



The fractions of variance explained by component in the middle panel of Fig. 5 indicate how the individual melting layer tops can be distinguished; melting layer tops with less spatial variability (two of the three Payerne cases) have most of their variance explained by the first component and then equal amounts of variance explained by all of the subsequent components. This means that the corresponding melting layer tops either vary only at the largest spatial scale or at even larger scales not resolved

by the obtained spatial frequencies. Melting layer tops which are spatially more variable have higher contributions to the total variance from smaller spatial scales, though here as well some leakage to the neighbouring frequencies is possible. It must be borne in mind that the larger spatial lags also have a higher associated uncertainty; the first component for example, is based on only one realisation of the series. Therefore, as an example, the ability of the first ten components to reconstitute a single melting layer top is given in Fig. 6. These first ten components represent 50% of the total variance of this signal as indicated

by the dashed lines in the middle panel of Fig. 5. The high spikes in the original melting layer (green line in the top panel) are artefacts from the melting layer detection algorithm, and some beam effects can also be observed at further distances (around 7000 m distance from the radar). Nevertheless, the series are rather well reconstituted by the first ten components, giving credibility to the accuracy of the method. Furthermore, for the comparison of the data from the two campaigns the individual fractions of variance explained by component have been regrouped into box plots in order to account for the uncertainties at

the larger spatial lags.

[FIGURE 6 about here]

The effects of de-trending and tapering the data before performing the Fourier transform are shown for the melting layer tops of both campaigns and for the artificial melting layer in Fig. 7. De-trending reduces the amount of variance explained by the first component as it decreases the amount of red noise; a linear trend acts as an infinite wavelength wave which manifests

itself with noise at low frequencies. As mentioned, some of the melting layers may still have signals which are longer than 10 kilometres. The fact that these have been truncated at a set distance may also result in some additional red noise at the lower frequencies. However, the fact that the first component does not always explain most of the variance and the ability of the components to reconstitute the original melting layer indicates that red noise does not dominate the fractions of variance explained by component. With the exception of the echo tops, where slopes were slightly higher, the values of the trends that

have been subtracted from the data were very similar for all datasets and were for the melting layer tops between -0.2 and 0.2; more than 50% of these remained within the -0.1 and 0.1 limits.

[FIGURE 7 about here]

## 4.2 Melting layer statistics

After running the melting layer detection algorithm on the retained scans, the outputs and derived variables as illustrated in Fig.

4 could be computed and descriptive statistics were calculated using the data over the entire scan. The resulting distributions of the melting layer tops, bottoms and depths, the highest reflectivity value and the lowest copolar correlation coefficient value within the melting layer and the height difference between these two values for all datasets are given in Fig. 8. A summary





of these statistics including statistics of other polarimetric variables is given in Table 2. For comparability with the Fig. 8, the statistics in Table 2 have been calculated on the log-transformed values. The comparison between DX50 values and MXPol values in Payerne allows to evaluate to what extent the observed differences in the statistics may be attributed to the different radar systems. The melting layer tops and bottoms display a bimodal distribution in the Valais with an approximately 400

metre shift to the lower values due to the lower zero degrees isotherm in this region and season. The distributions of the heights of the melting layer tops and bottoms for the DX50 and MXPol radars (for the same events) are comparable, with some second-order differences which could be explained by the different locations of the radar. Distributions of the melting layer depths are slightly skewed with a mode around 300 metres. The histograms of the melting layer tops, bottoms and depths show remarkable coherence in that the height distributions on the same location are very similar, that there is a shift towards the

lower heights for the Valais data and that the melting layer depths remain the same between locations. The bimodality of the Valais data may be explained by the exceptional character of the 2016-2017 winter season with unseasonally high temperatures and perhaps the inclusion of data from early spring notwithstanding our careful selection of the data. Nevertheless, they are comparable to the results from Cluckie et al. (2000) in Salford England, where a bimodal distribution of melting layer heights with peaks at 650 metres and 1850 metres was found. The melting layer tops are also within the limits of the values found

by Fabry et al. (1994b) in Montreal, which ranged between 200 and 3800 metres in spring and 0 and 3900 metres in winter. The observed melting layer depths are thicker than those observed by Cluckie et al. (2000) but comparable to the depth ranges reported by Fabry et al. (1994b). Similarly to the results in Cluckie et al. (2000) and Wolfensberger et al. (2016) melting layer thickness seems to be independent of season, climate or topography. It thus appears that at least in these datasets there is no indication of a relationship between melting layer thickness and melting layer height as suggested in Fabry et al. (1994b).

Instead, as found by Fabry and Zawadzki (1995) it is more likely that there is some positive correlation between the melting layer thickness and the reflectivity of rain below the melting layer. Though distributions of the reflectivity values also seem very similar between the datasets, the Valais dataset is slightly shifted towards higher reflectivity values. This may be an effect of sampling (as the Payerne dataset is smaller) or due to the presence of higher precipitation intensities in the Valais dataset, but is also conform with the results from Wolfensberger et al. (2016) where a shift between the Payerne data and the Davos

data (also a mountainous area) can be observed.

[FIGURE 8 about here]

The distributions of the lowest values of the copolar correlation coefficient in the melting layer show much lower values for the DX50 than for the MXPol radar; this is a known deviation for this radar. The Valais dataset, like the Davos dataset in Wolfensberger et al. (2016) shows a larger presence of lower $\rho_{hv}$ values within and above the melting layer but similar overall

distributions. Lower $\rho_{hv}$ values could be associated with enhanced riming and depositional growth above the melting layer. These processes result in the presence of a higher variety of particle types and can be expected to be the dominant growth mechanisms in a winter orographic environment and in situations with a low melting layer (Colle and Zeng, 2004a, b; Colle et al., 2005a, b; Stoelinga et al., 2013; Schneebeli et al., 2013). Finally, the distance between the reflectivity maximum and the



copolar correlation coefficient minimum shows a similar distribution across all datasets.

[TABLE 2 about here]

## 4.3 Spatial variability

The fractions of variance explained by component have been calculated for each separate RHI scan; which acts as a type of normalisation since each fraction represents the fraction of the total variability of the detected melting layer in that scan. Because the detected melting layers do not all have the same length, the fractions of variance explained by component have been regrouped into spatial lags based on their corresponding frequency values. As demonstrated in Eq. (4), the fractions of variance explained by component only give values of one side of the Fourier spectrum, in order for the fractions to sum to

1, the spectra should be folded (i.e. multiplied by 2). Figure 9 shows the box plots for the melting layer tops; both the DX50 and the MXPol data from the Payerne campaign show higher fractions of variability explained for the first components (larger spatial lags) and an exponential decline of the fractions of variance explained towards the smaller spatial lags. The box plots for the Valais data on the other hand, display a much less pronounced decline in these fractions towards the smaller spatial lags indicating, on average, a higher relative importance of the variability at smaller spatial scales in the Alpine winter environment.

Random sub-sampling of the Valais dataset indicated that these results are robust also for a smaller number (460) of scans. And analysis of the spatial variability at the event scale showed similar behaviour of the components across events. Moreover, the first ten components shown in the box plots explain, on average and for the folded spectra, 43%, 42% and 36% of the total variance for the DX50, MXPol (Payerne) and MXPol (Valais) data respectively. The difference between the Valais and Payerne datasets decreases when the first 20 components are considered to 54%, 53% and 50%.

20                                  [FIGURE 9 about here]

The levelling out of the variability after the fifth component (the subsequent component corresponds to spatial scales of 2000 - 1500 metres) is comparable to the findings in Wolfensberger et al. (2016) who noted that the variogram of the melting layer tops reaches decorrelation distance at around 1500 metres. As can be seen in the box plots in Fig. 9 the intra campaign variability remains quite large. In fact, the first component is not always the most important component in terms of fractions of variance explained. And subsequent components are more often the most important component in the Valais data then in the Payerne

datasets (Fig. 10). This intra campaign variability can be separated into inter event variability and intra event variability. Figure 11 shows a parallel coordinates plot of the fractions of variance explained for the binned spatial scales (on the vertical parallel lines) for each individual RHI scan (coloured lines) of the separate events (grouped on the first vertical line). In the left panels of Fig. 11 10% of the explained variability of the second component has been highlighted to illustrate the inter-event variability. In the right panel, a single event has been highlighted such that the variability of the values for the components in a single

event (intra-event variability) becomes evident. The inter event variability of the fractions of variance explained by component appears to be larger for the Payerne data (upper middle panel in Fig. 11), as is illustrated by the many different colours in the



selected 10% of the second component. This may well be due to a sampling effect as the dataset is much smaller and thus the weight of the individual scans is much more important. For the Valais data, the intra-event variability and the inter event variability appear more similar. The high intra-event variability in the fractions of variance explained by component suggests that the melting layer variability is not necessarily consistent for similar synoptic conditions. Notwithstanding this inter and

intra event variability, the box plots of the fractions of variance explained by component at the event scale indicate a typical behaviour namely that the variability in space at the larger scales is always smaller in the Valais data, and in the Payerne data most of the total variance is always explained by the first component (i.e. the larger scales).

[FIGURE 10 about here]

The fractions of variance explained by the melting layer bottoms are more similar for both campaigns and showed higher values for the larger spatial lags in the Valais data than for the melting layer tops. This has been commented on in Wolfensberger et al. (2016) and is thought to be related to the fact that the detected melting layer bottom is smoother than the tops because the detection of the melting layer top is solely dependent on reflectivity which is more influenced by large hydrometeors, while the detection of the bottom depends on both reflectivity and the copolar cross-correlation coefficient. The box plots of the

fractions of variance explained by component of the melting layer depths (thickness) show very little spatial variability and indicate the opposite behaviour as compared to the melting layer tops. For the Valais data, the larger spatial lags explain most of the variance, whereas for the Payerne data, the 20 - 15 km and the 15-10 km (15-10 km and 10-5 km for the $\rho_{hv}$ minimum) lags are equally important. This may be related to the presence of more convective pockets in the Payerne dataset. Overall, the statistics are more similar for the two locations (and for the three radars) for the melting layer depths than for the melting layer

tops, indicating that the variability in the melting layer depth is more consistent over flat areas and complex topography.

[FIGURE 11 about here]

**4.4    Spatio-temporal coherence of the melting layer variability**

For the analysis of the temporal variability, a number of events have been selected based on their duration (at least 24 hours of

quasi continuous precipitation). As such, from the Payerne campaign 4 events with a total duration of 26.5 hours were selected, and from the Valais campaign 4 events with a total duration of 94 hours were selected. In order to increase the temporal resolution, the data extracted at vertical incidence from the RHI scans were added to the vertical PPI scans, creating blended RHI/PPI time series of at least 1.66 minute resolution. The melting layer detection algorithm was then applied to the temporal series which were de-trended over the duration of the entire event, thus preserving the within-event variability. The removed absolute

trends varied depending on the length of the time series; the highest removed trend was 112 metres/hour for the shortest event and the lowest removed trend was 3 metres/hour for the longest event. The 112 metres/hour trend may seem large, but has been related to the passage of a small occluded front, and remains lower than the observed height change in a time series from Fabry





et al. (1994b), who recorded a change of 1.5 km in 3 hours. From the de-trended data, periods of one hour were selected with a sliding window of 12 minutes. The lengths of the sub selected time series were roughly the same, but they have nevertheless been re-binned into set time lags. Figure 12 shows the resulting box plots of the fractions of variance explained by component for both the spatial components (in green) and the temporal components (in red and orange) for the two campaigns. Figure 12

suggests that in Payerne, the spatial and temporal variabilities are very similar at the investigated scales. The slightly lower values for the temporal data may be attributed to the fact that it was more difficult to ascertain continuous precipitation in the temporal data and discontinuities have the effect of diminishing the spectral slope and as such also the fractions of variance explained for the larger scale components (De Montera et al., 2009; Verrier et al., 2011; Rysman et al., 2013). Still, variability observed at spatial scales of 20 to 15 kilometres is very similar to the variability observed at the 1 hour scale. Indeed, consid-

ering that the average wind speed measured by a meteorological station on site during the time periods of the selected events was 11.0 km/h ± 3.7 km/h, it may well be that the air mass scanned by the RHI scans up to 10 km from the radar was very similar to the air mass scanned above the radar within the hour. This is also comparable to the findings of Rysman et al. (2013), who found similar spectral slopes for the 20-45 minutes temporal range and the 7-20 km spatial range for summer months in the Mediterranean region. The authors related the spectral slope at these temporal ranges to the expected value for velocity

within a turbulent flow, indicating that at these scales rain is driven by turbulence. Whereas in the Mediterranean case the spectral slope for the 20-45 minute time lag was the same for all months (except during fall), the Payerne and Valais datasets show more discrepancies. For the Valais data at the one hour scale (red box plots in the right panel of Fig. 12), the larger time lags seem to be responsible for about twice as much of the temporal variability than the larger spatial lags. One the one hand, this could be consistent with the influence of the more variable small-scale topography in the Valais affecting the RHI scans.

On the other hand, the wind speeds during the Valais events were also much more variable with an average of 4.2 km/h and a standard deviation of 5.3 km/h. The slower wind speeds and longer time series of the Valais events justifies an analysis at longer temporal scales. The right panel in Fig. 12 also shows the box plots for the temporal components for spatial scales up to 2 (dark orange) and three (orange) hours. Though some differences with the spatial components can still be observed, the temporal components are much more similar at the 2 hourly and 3 hourly scales. The difficulty of finding spatio-temporal

coherence in the Valais data may also partly be explained by the wintertime results from Rysman et al. (2013) where spectral slopes showed that a unique scaling regime characterised the rainfall scaling behaviour from 3 to 70 km scales (and from 5 minute to 3 hourly scales), meaning that the fractions of variance explained by component can be expected to be very similar at these scales.

In order to assess the influence of the topography, a similar analysis has been applied by extracting data from a digital eleva-tion model (Jarvis et al., 2008) (3 arc second (∼90m horizontal resolution), 16 metre vertical accuracy (Smith, 2003)) sampled at a 25 metres resolution along the transect of the dominant wind directions (200 - 250° for Payerne, 100 - 200° and 300 - 350° for Valais) and for distances corresponding to the detected melting layer lengths observed during the two campaigns. While for Payerne a clear dominance could be observed at the 20 - 15 km scale, for the Valais data, the 10 - 5 km scale was almost

equally important as the 20 - 15 km scale. Though these results indicate a higher importance of the smaller scale topography





in the dominant wind directions in the Valais, it is difficult to link these results to the melting layer variability due to the many other factors (i.e. wind speeds and hydrometeor fall velocities) that may play a role.

[FIGURE 12 about here]

**5    Conclusions**

This study presented the characterisation and quantification of the spatial variability of the melting layer at different scales using a method based on the Fourier transform. It is demonstrated that the proposed method is able to accurately describe the variability in individual melting layers and that it constitutes a useful basis for comparison of the variability of the observed melting layer at different spatial scales in different regions or from different data sources. The method has been applied to data

from measurement campaigns conducted in two very different though highly representative regions in Switzerland, namely on the relatively flat Swiss plateau during summer and in a large inner Alpine valley in the Swiss Alps in winter. The descriptive global statistics of the melting layer tops, bottoms and depths have been found to display remarkable coherence; with a seasonal shift in the distributions of the heights while distributions of the melting layer thickness remained the same independent of season or location. The values and distributions found in this study are consistent with those found in previous studies at

other locations. However, the performed Fourier analysis of the spatial variability of the melting layer tops indicated a higher importance of variability at smaller spatial scales in the case of the Alpine environment, possibly related to the influence of the small-scale topography. The investigation of the variability of the topography in the dominant wind directions of the considered regions also suggested a larger importance of the small scales in the Valais region, but will require further research in order to establish a more direct link. According to the results of this study, there is little difference in the spatial variability

of the melting layer thickness in the two regions suggesting that it is less affected by topography. Finally, the method was also applied to time series of the melting layer height of sufficiently long events in order to study the spatio-temporal coherence of the melting layer variability. For the Swiss plateau, the variability at the 1 hourly temporal scale corresponded well with the spatial variability at the 15-20 km scales, which is also consistent with results from other studies. The average wind speeds during these events varied little around 11 km/h suggesting that the hypothesis of pure advection is quite valid for this region

and during spring time. Due to more variable wind speeds and directions, the presence of small-scale topography and the possible scale invariance of wintertime precipitation conditions, it was more difficult to relate the scales of spatial and temporal variability in the Alpine environment. If any, it is possible that the spatio-temporal coherence in this region occurs at larger scales than could be measured with an X-band radar.

It should be noted that the results of this study are restricted to very specific locations and conditions as well as to temporal

scales of up to 3 hours and spatial scales of 20 km and less. Nevertheless, the presented results indicate that for some regions the descriptive global statistics of the melting layer height may hide some important spatial variability of the melting layer. Current operational vertical profile correction techniques still assume spatial homogeneity of the melting layer, and the results of this





study further indicate the necessity of a correction technique which takes this variability into account, and give an indication of the relative contributions of various scales.

*Code and data availability.* The py-ART Radar Toolkit by ARM-DOE which was used within the context of this paper is available at: http://arm-doe.github.io/pyart/. Datasets acquired by the MXPol X-band radar can be made available upon request to the authors. For data

5    acquired by the DX50 X-band radar, contact the authors affiliated with MeteoSwiss.

*Author contributions.* FH and AB developed the concept of the paper, FH performed the analyses and FH, AB, MG and UG interpreted the results. FH with contributions of all authors, prepared the manuscript.

*Competing interests.* The authors declare that they have no conflict of interest.

*Acknowledgements.* The authors would like to thank their colleagues at LTE (EPFL) and MeteoSwiss for their useful suggestions and

10   support. In particular, we are indebted to Daniel Wolfensberger for his support with the melting layer detection algorithm, as well as to Jacopo Grazioli and Jordi Figueras i Ventura for their support with the processing of the data and their roles in the organization of the PaRaDIso (and Valais) campaigns. We would also like to thank Loris Foresti and Nikola Besic for their insightful comments and discussions.





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



## List of Figures

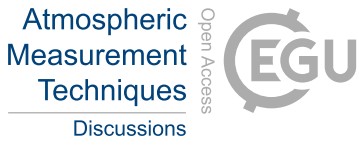

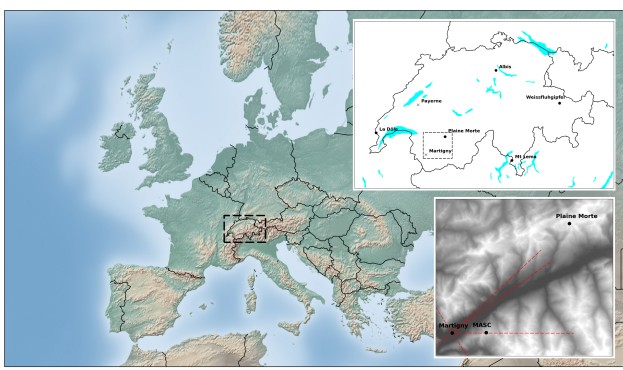

**Figure 1.** Locations of the study areas (Payerne and Martigny) within Switzerland (top left panel) and the scan directions for the RHI scans during the Valais campaign (lower left panel).





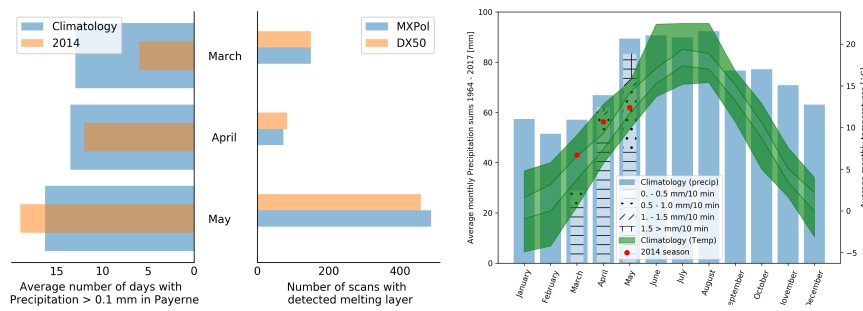

**Figure 2.** Meteorological conditions in Payerne during the campaign compared to climatology; with the average number of days with precipitation for each month (left panel), the number of retained scans with a detected melting layer (middle panel) and a climatogram (right panel).





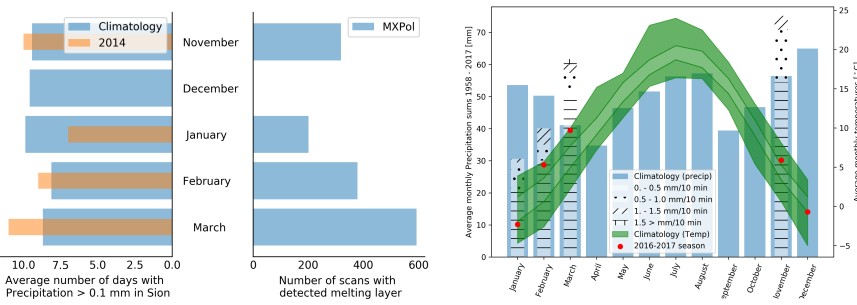

**Figure 3.** Meteorological conditions in Sion (~20 km from MXPol) during the campaign compared to climatology; with the average number of days with precipitation for each month (left panel), the number of retained scans with a detected melting layer (middle panel) and a climatogram (right panel).





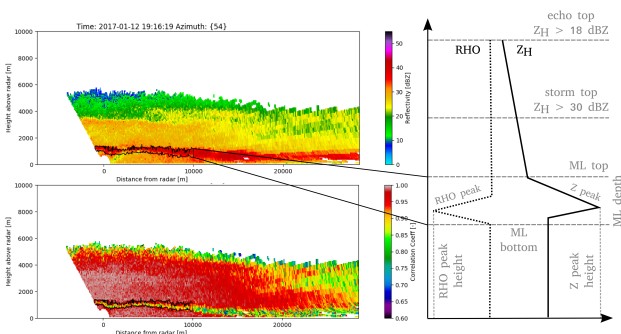

**Figure 4.** Melting layer detection algorithm output superimposed on the reflectivity values (left upper panel) and on copolar correlation coefficient values (left lower panel) and the extracted and derived variables (right) indicated for an idealised vertical profile of reflectivity (VPR).



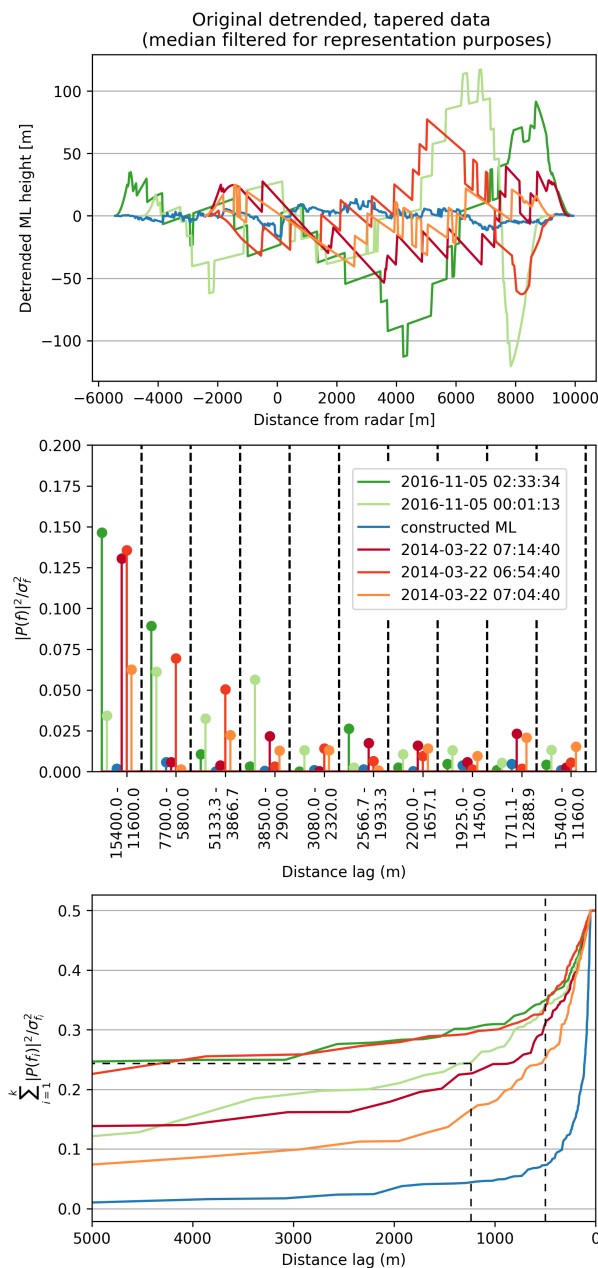

**Figure 5.** Examples of observed melting layer tops (top panel), their corresponding fractions of explained variance by component (middle panel) and the cumulated fractions of explained variance (bottom panel) from the Valais campaign (green), from the Payerne campaign (red) and for a constructed melting layer consisting of white noise with a drift (blue).





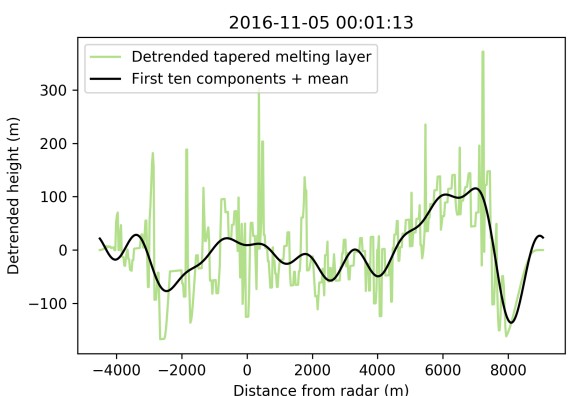

**Figure 6.** Original melting layer and the reconstituted melting layer from the first ten components.





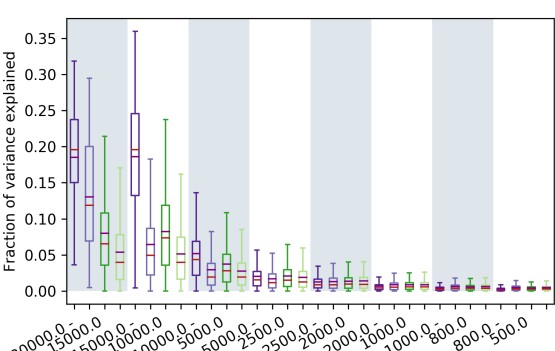

**Figure 7.** Box plots of the original melting layer tops (on the left) and the de-trended and tapered melting layer tops (on the right), for the Payerne campaign (in purple) and the Valais campaign (in green).



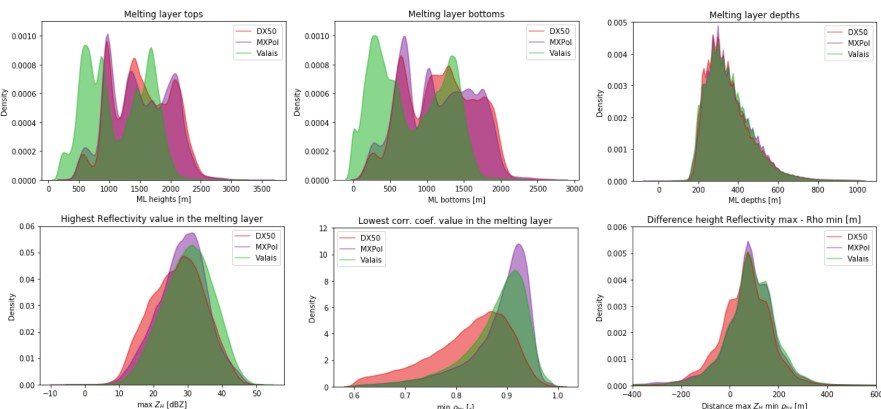

**Figure 8.** Distributions of the characteristic melting layer variables.




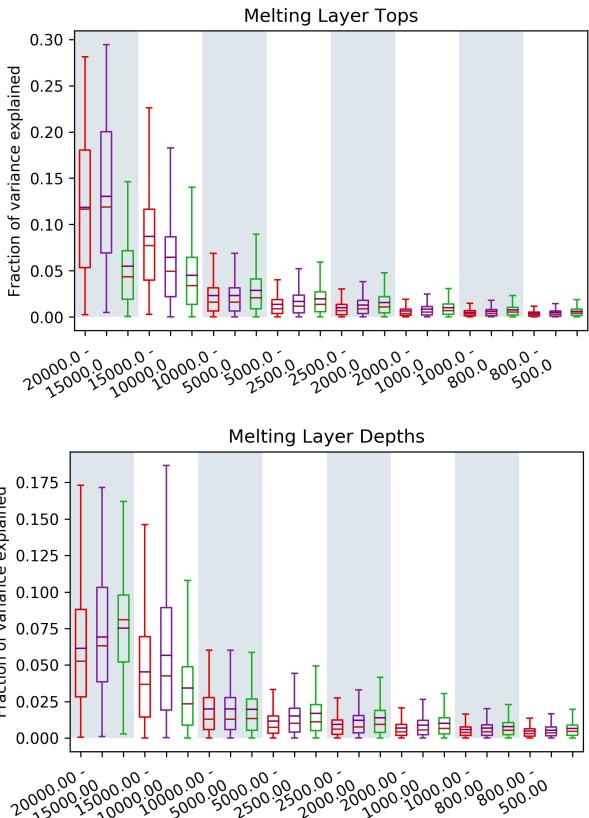

**Figure 9.** Fractions of variance explained by component for the melting layer tops (top panel) and the melting layer depths (bottom panel) for the DX50 (red), MXPol in Payerne (purple) and MXPol in the Valais (green) (fractions of individual melting layers have been binned based on their corresponding frequency values).





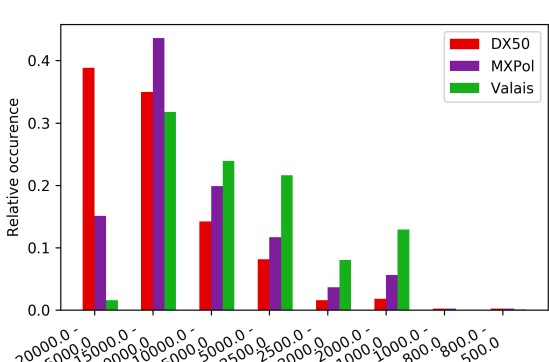

**Figure 10.** Distribution of the binned spatial component with the highest fraction of explained variance per scan for the DX50 (red), MXPol in Payerne (purple) and MXPol in the Valais (green).




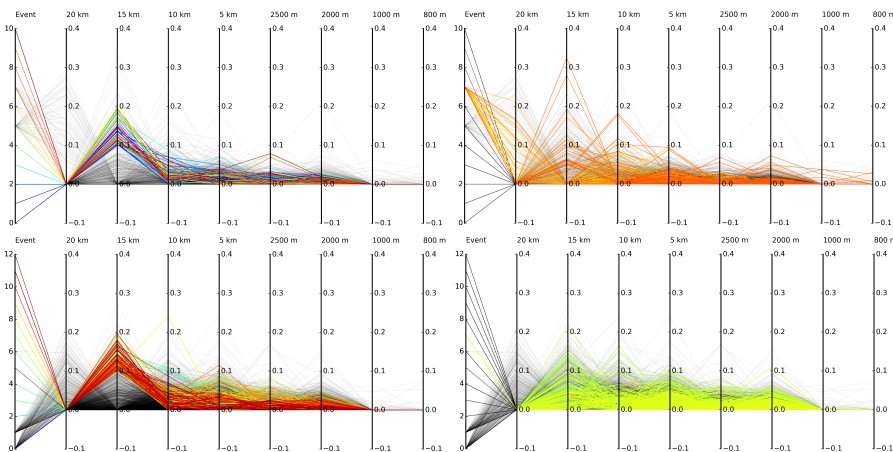

**Figure 11.** Parallel coordinates plots illustrating the intra en inter event variability of the fractions of variance explained by component of the melting layer tops, for the MXPol in Payerne (top panels) and the MXPol in the Valais (bottom panels); highlighted for 10% of the explained variance for the second component (left panel), and highlighted for a single event (right panel).





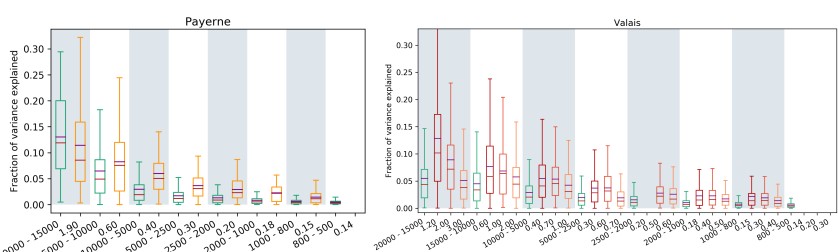

**Figure 12.** Box plots of the fractions of variance explained by component (binned) for the melting layer tops of the selected events for MXPol data in Payerne (left panel) and MXPol data in the Valais (right panel). Spatial components are in green and temporal components are in orange for the Payerne data and in hues of red for the Valais data to distinguish the different time periods of 1, 2 and 3 hours.



## List of Tables



**Table 1.** Characteristics of the MXPol radar and scanning strategy during the Valais campaign.

| Radar parameters | |
| --- | --- |
| Wavelength | 3.2 cm |
| Diameter | 185 cm, **183 cm** |
| Range | 35 km |
| 3dB beamwidth | 1.43° / **1.27°**[3] |
| Peak power | 7.50 kW per channel |
| Radial Resolution | 30, **75** m |
| Polarization | Simultaneous H-V |
| Scan strategy | |
| 3 Range Height Indicator (RHI) | 47, 90, 147° azimuth |
| 1 emispheric RHI | 90° azimuth |
| 1 PPI for $Z_{DR}$ calibration | 90° elevation |
| Scanning | Dual Pulse Pair mode |
| | FFT mode |
| Sequence duration | 3 minutes 45 seconds |



**Table 2.** Statistics of the polarimetric variables related to the melting layer for the DX50 and MXPol in the Payerne and Valais campaigns.

| Variable | Statistic | DX50 (pay) | MXPol (pay) | MXPol (val) |
|---|---|---|---|---|
| $Z_H$ | Mean | 22.19 | 23.14 | 22.84 |
| | St.Dev | 8.72 | 8.45 | 8.82 |
| | Q10 | 11.5 | 11.90 | 11.77 |
| | Q50 | 22.0 | 23.25 | 22.36 |
| | Q90 | 34.0 | 34.0 | 35.02 |
| $Z_{H\text{peak}}$ | Mean | 27.18 | 28.49 | 30.18 |
| | St.Dev | 7.73 | 7.11 | 7.23 |
| | Q10 | 16.5 | 18.83 | 20.44 |
| | Q50 | 27.5 | 28.95 | 30.47 |
| | Q90 | 37.5 | 37.15 | 39.49 |
| $Z_{DR}$ | Mean | 1.04 | 0.6 | 1.11 |
| | St.Dev | 1.13 | 0.86 | 1.06 |
| | Q10 | -0.13 | -0.3 | 0.05 |
| | Q50 | 0.90 | 0.47 | 0.92 |
| | Q90 | 2.47 | 1.76 | 2.39 |
| $\rho_{hv}$ | Mean | 0.9125 | 0.9256 | 0.9301 |
| | St.Dev | 0.0727 | 0.0618 | 0.0595 |
| | Q10 | 0.8111 | 0.8457 | 0.8506 |
| | Q50 | 0.9331 | 0.9421 | 0.9465 |
| | Q90 | 0.9803 | 0.9841 | 0.9874 |
| $\rho_{hv\text{min}}$ | Mean | 0.8245 | 0.8854 | 0.8785 |
| | St.Dev | 0.0796 | 0.0633 | 0.0637 |
| | Q10 | 0.7047 | 0.7971 | 0.7912 |
| | Q50 | 0.8386 | 0.9039 | 0.8937 |
| | Q90 | 0.9173 | 0.9437 | 0.9430 |
| $K_{dp}$ | Mean | 0.06 | 0.098 | 0.099 |
| | St.Dev | 0.5 | 0.2 | 0.2 |
| | Q10 | -0.5 | -0.13 | -0.045 |
| | Q50 | 0.05 | 0.09 | 0.03 |
| | Q90 | 0.61 | 0.32 | 0.33 |