# Peer review of "Characterisation of the melting layer variability in an Alpine valley based on polarimetric X-band radar scans"

_Atmospheric Measurement Techniques, 2018_

## Referee Comment (RC1) · Anonymous Referee #1 · 19 Jun 2018

The paper deals with an accurate method to characterise the spatial and temporal variability of the melting layer using the spectral analysis. The method has been applied to a large dataset of high resolution X-band polarimetric radar data from two measurement campaign in Switzerland. The main merit of the manuscript is to apply for the first time a method based on the Fourier Transform to investigate the variability in space and time of the melting layer. The methods and results of the paper are well described and discussed but in my opinion the conclusions are not adequately supported by the analysis.

Major comments:

(1) It is not clear how the effects of the topography have been related to the spectral analysis. We know that the spatial and temporal behaviour of the melting layer depends not only on the orographic context but also on the temperature, humidity and microphysical processes. At the end of Section 4.4 an analysis from a digital elevation model (DEM) has been mentioned but not figures or quantitative measurements are shown. To enhance the conclusions, I suggest to show one figure/table by comparing the DEM (or a statistic clutter map) with the spectral analysis of the melting layer.

(2) I suggest to strongly modify Figure 11 because it is hard to read.

Minor comments:

(1) Figures 5(b), 7, 9, 10 and 12: the x-label is missing.

(2) Figure 7: the legend is wrong.

---

## Author Comment (AC1) · 31 Jul 2018

We thank the anonymous referee for his insightful and helpful comments. Our responses are numbered in accordance with the referee's remarks:

Major comments:

(1) Several analyses were performed in this respect, including the analysis based on data extracted from a DEM along the transects of the RHI scans. The results of this analysis are now displayed in Figure 13, and described on page 15 lines 1-7.

(2) The figure has been modified to show both selections in a single figure so that it

now comprises two panels instead of four. The text and the axes labels have also been enlarged. The figure legend and the text on page 12 lines 27-32 have been modified accordingly.

Minor comments:

(1) X-labels have been inserted for these figures and in accordance with the comment from Referee #2, changed to "Spatial scales" for the other figures. To enhance readability, the scales are now given in km rather than m for all figures except figure 5.

(2) The previous label of Figure 7 was indeed confusing. A legend has been added within the figure and the figure label has been changed to: "Box plots illustrating the effects of median filtering, de-trending and tapering on the original melting layer tops." Which should facilitate the interpretation of the figure.

---

## Author Comment (AC2) · 31 Jul 2018

The comment was uploaded in the form of a supplement:
https://www.atmos-meas-tech-discuss.net/amt-2018-145/amt-2018-145-AC2-supplement.zip

---

## Author Response (AR1)

**Characterisation of the melting layer variability in an Alpine valley based on polarimetric X-band radar scans**

Point-by-point response to all referee comments

Floor van den Heuvel, Marco Gabella, Urs Germann and Alexis Berne

**Anonymous Referee #1**

The paper deals with an accurate method to characterise the spatial and temporal variability of the melting layer using the spectral analysis. The method has been applied to a large dataset of high resolution X-band polarimetric radar data from two measurement campaigns in Switzerland. The main merit of the manuscript is to apply for the first time a method based on the Fourier Transform to investigate the variability in space and time of the melting layer. The methods and results of the paper are well described and discussed but in my opinion the conclusions are not adequately supported by the analysis.

Major comments:

(1.1) It is not clear how the effects of the topography have been related to the spectral analysis. We know that the spatial and temporal behaviour of the melting layer depends not only on the orographic context but also on the temperature, humidity and microphysical processes. At the end of Section 4.4 an analysis from a digital elevation model (DEM) has been mentioned but no figures or quantitative measurements are shown. To enhance the conclusions, I suggest to show one figure/table by comparing the DEM (or a statistic clutter map) with the spectral analysis of the melting layer.

(1.2) Several analyses were performed in this respect, including the analysis based on data extracted from a DEM along the transects of the RHI scans.

(1.3) The results of this analysis are now displayed in Figure 13, and described on page 15 lines 1-8.

(2.1) I suggest to strongly modify Figure 11 because it is hard to read.

(2.3) The figure has been modified to show both selections in a single figure so that it now comprises two panels instead of four. The text and the axes labels have also been enlarged. The figure legend and the text on page 12 lines 27-32 have been modified accordingly.

Minor comments:

(1.1) Figures 5(b), 7, 9, 10 and 12: the x-label is missing.

(1.3) X-labels have been inserted for these figures and in accordance with the comment from Referee #2, changed to "Spatial scales" for the other figures. To enhance readability, the scales are now given in km rather than m for all figures except figure 5.

(2.1) Figure 7: the legend is wrong.

(2.2) The previous label of Figure 7 was indeed confusing.

(2.3) A legend has been added within the figure and the figure label has been changed to: "Box plots illustrating the effects of median filtering, de-trending and tapering on the original melting layer tops." Which should facilitate the interpretation of the figure.

**Anonymous Referee #2**

General Comments

This paper presents an analysis of small scale variability mainly of melting layer height in complex terrain (the region of Alps) using close range data from RHI scans acquired with X-band polarimetric radars. The analysis focuses on the identification of the larger spatial scale which can explain most of the variability of the melting layer height and less details are showed for other characteristics of the melting layer. The paper shows the significance of accounting for the spatial variability of melting layer for more accurate operational radar products in complex terrain. Some clarifications and modifications are needed before final publication.

Specific Comments

(1.1) Page 4, line 18: In PPI scans azimuth scanning is assumed. Consider using a term like "vertical beam recording" instead of "vertical PPI scan", unless the antenna is actually rotated while pointing in the vertical direction.

(1.2) The antenna was actually rotated while pointing in the vertical direction, which is why the term "vertical PPI scan" has been used here.

(1.3) The words "(rotating 360 degrees)" have been added to this line to emphasize this.

(2.1) Page 5, lines 15-20: Some details on DPP and FFT modes and ZDR calibration could be given. Also, beam width and antenna rotation rate/dwell (rays) averaging should be given for an indication of vertical resolution in RHI scans. Was an interpolation of RHI scans in a regular vertical grid performed before analysis?

(2.2) Information on the DPP and FFT modes have been added to Table 1.

(2.3) The terms "vertically pointing PPI scan (360 degrees rotation)" and "ZDR calibration scan" refer to the same scan. Since ZDR is neither used within the context of this study nor for the melting layer detection algorithm, it was not considered relevant to give more information on ZDR calibration. To clarify this, on Page 4, line 18 "(rotating 360 degrees)" has been added and on Page 5, line 16 "ZDR calibration scan" has now been changed to "vertical PPI scan (rotating 360 degrees)" so that it is similar to section 2.1.

The beam width and antenna rotation rate are given in Table 1.

The melting layer detection algorithm requires that the RHI scans should be interpolated on a Cartesian grid. Best performance was obtained for a Cartesian grid of 25 m (Wolfensberger et al. 2016). In order to clarify this, the last line of section 3.1 (Page 6, line 27) has been adapted to "Otherwise, the default settings found to be optimal and described by Wolfensberger (2016) were used, including interpolation of the RHI scans on a 25 m resolution Cartesian grid."

(3.1) Page 6, line 18: A comparison of RHI melting layer detection with common in space and time

detections in near horizontal PPI scans could be useful for spatial variability for melting layer.

(3.2) Unfortunately, for the Valais data only very few near horizontal PPI scans are available and only at the beginning of the campaign. Because the radar was located in a deep Alpine valley, any non-vertical PPI scan was for a considerable part blocked by mountains, and thus not considered very useful.

(4.1) Page 6, line 24: It should be mentioned that the 10 km short range was used for analysis to e.g. avoid beam broadening effects (assuming even smaller non-uniform beam filling effects). This can be seen in Fig. 4. In addition, in that figure there is a significant rapid decrease of melting layer at distances larger than 10 km. Is there an explanation for this observation, e.g. proximity to the edge of the rain cloud?

(4.2) This is mentioned on page 6, line 24: "In order to limit the effects of beam broadening, the melting layer detection algorithm has been set to detect up to a maximum distance of 10 km from the radar".

The trapping of cold air in the valley may happen regularly in the Valais, and could explain the bending of the melting layer towards the ground near Sion. For the case in Fig. 4 for example, we have observed a cold gradient between Evionnaz (~10 km West of the radar) and Sion (at the same altitude, ~20 km to the East and in the direction of the RHI scan), of about -0.6 °C.

(4.3) This explanation has been added to the label in Fig.4: "The bending of the melting layer towards the ground is probably related to the trapping of cold air in the valley and the observed negative temperature gradient towards the East (in the direction of the scan)."

(5.1) Pages 7-8: The details (equations) of FFT are well known and could be omitted if they are not really useful.

(5.2) The FFT equation (former equation 2) has been removed. The other equations illustrate the relation between spectral slopes and fractions of variance explained by component, and are therefore considered helpful for understanding the results of this study and for the comparison with other studies.

(5.3) The text on Pages 7-8 and Lines 19 – 6 has been adapted accordingly.

(6.1) Page 9, line 1: Spatial lags probably correspond to wavelengths? In this case, the term "spatial scales" is probably closer to this meaning.

(6.3) The term "Spatial lags" has been changed to "spatial scales" in the figures as well as in text.

(7.1) Page 9, Fig. 5: According to this figure, the melting layer detections are not consecutive with relatively large range gaps where linear interpolation (rather than median filter) is done. Some explanation for these missing detections could be given like a quality check of melting layer detection. This could lead to significant errors in spectral analysis of small "lags". If the raw detections are as shown in Fig. 6, the abrupt changes (spikes of more than 100m) should be "noise" in the detection algorithm and it should be removed (filtered) before spectral analysis.

(7.2) The interpolation of the melting layer detections is part of the melting layer detection algorithm as mentioned on page 6, line 25: "holes in the detected melting layer tops and bottoms were interpolated up to a maximum length of 1500 metres". However, in Fig. 5 it is really the median filtering (for representational purposes only) which suggests these relatively large gaps. For

the quality check of the melting layer detection we refer to the paper by Wolfensberger et al. 2016.

The observed spikes (of more than 100 m) occur at the 25- 50 metre horizontal scales, which are not included in the first ten components analyzed in this study. The referee correctly noted that these abrupt changes may influence the spectral slope. However the effect on the fractions of variance explained by component remains minor, as is also illustrated by Fig 7. on Page 10 to which the influence of a hypothetical median filtering before performing the detrending and bell-tapering has now been added.

(7.3) Fig 7. on Page 10 has been adapted to include the effects on the fractions of variance explained by component when performing a median filtering of the melting layer variables before performing the analysis. Lines 27 - 30 on page 10 have also been adapted: "Artefacts from the melting layer detection algorithm or noise from the original measurement may have some influence on the spectral slopes, which is why Fig. 7 also shows the effects of performing an additional median filtering of the melting layers before de-trending and tapering. It appears that the effect of median filtering on the fractions of variance explained by component is minor, and that de-trending and tapering of the melting layers is sufficient."

(8.1) Page 10, Fig. 7: Spatial "lags" up to about 20 km are shown, but even if the RHI scans are from 0 to 180 degrees antenna elevation (which is not likely due to terrain, like it is shown in Fig. 4) the 10 km range should give spectral information for "lags" up to 10 km (or less) similar to the limitation of Nyquist frequency in FFT.

(8.2) In fact, both in Payerne and in Valais, the hemispheric scan was performed from 0 – 180 degrees, Thus, 20 km ranges are possible and observed in some cases (though dependent on ML length and terrain, as observed in Figure 4). In the Valais, as mentioned on page 5, line 16, the scan strategy changed during the campaign. Figure 4 comes from the second scan strategy which was more hindered in the 0-23 degrees elevation range by terrain. However, many of the RHI scans included in this analysis come from the first scan strategy, during which measurements up to 20 km were possible. As can be observed in Figure 11 Page 13 both campaigns have a comparable number of scans within these spatial scales. Admittedly, 20 km is the absolute upper limit, which is why the bin is defined as 15-20 km, so that the longest series from both scan strategies and campaigns can be included in this group.

(8.3) To emphasize this, the following lines have been added:

Page 9, lines 21-22 : "The largest wavelengths (or spatial scales) correspond to distances of 20-15 km for melting layers which spanned almost the entire hemispheric scan."

Lines 12-15 Page 13 : "The coordinate plots also show that the larger spatial scales (20-15 km) are equally well represented in both campaigns even though for the Valais campaign these only occurred in the first two events because of a change in scan strategy which hindered the visibility in the 0-23 degrees elevations afterwards."

(9.1) Page 11, Fig. 8: The copolar correlation coefficient values from DX50 radar are too low. This may indicate e.g. some synchronization problem in H-V channels rather than a physical explanation. Thus, this could lead to significant errors in melting layer detection where the correlation coefficient is the critical parameter. This is probably indicated by the results like for the larger "lags" in Fig.10.

(9.2) Indeed, this is a known deviation for this radar (page 11, lines 33-34). The melting layer detection is based on gradients of Rhohv and ZH, which are scaled. As such, the absolute values

should not have an influence on the detection.

(9.3) This has now been added to lines 33-34 page 11: "[..] this is a known deviation for this radar, but does not affect the melting layer detection algorithm which is based on scaled gradients of Rhohv and ZH."

(10.1) Page 12, Fig. 11: The plots are too crowded by lines. Maybe some kind of standard deviation could be used instead of showing all lines.

(10.2) It is in the interest of showing the intra-event variability that all the lines are shown, since the box plots in Fig. 9 page 12 already give a measure of the spread of the entire dataset.

(10.3) Fig 11. Page 13. has been adapted so that now it only consists of 2 panels instead of 4, improving the readability.